# How Do Consumer Innovation Characteristics and Consumption Value Shape Users' Willingness to Buy Innovative Car Safety Seats?

**Li Jiang** [1,2] ⬤, **Mei Zhao** [1,2,*], **Hao Lin** [3] and **Lvyu Yang** [1,2]

1    CAS Key Laboratory of Mental Health, Institute of Psychology, Chinese Academy of Sciences, Beijing 100101, China
2    Department of Psychology, University of Chinese Academy of Sciences, Beijing 100049, China
3    Shanghai Woyoo Electronic Technology Co., Ltd., Shanghai 201112, China
*     Correspondence: zhaomei@psych.ac.cn; Tel.: +86-010-64879520

**Abstract:** The intelligent innovation of child safety seats has brought new impacts and challenges to the Chinese market. Researchers in the car seat industry have been focusing on industry regulations and the abuse of car seats, but there is a lack of consumer-centered research. This study is the first to combine two theories of consumer subject-specific innovation (DSI) and the theory of consumption value (TCV). This study explores how consumer innovations influence consumers' purchase of innovative child safety seats through perceived value. The proposed research model was evaluated using a partial least squares structural equation model, and data analysis revealed that the model had good model fit, reliability, and validity. Consumer product innovation has a significantly better impact on willingness to buy than consumer information innovation. In this study, in the relationship between consumers of information innovation and purchase intention in the automobile seat industry, a new kind of parallel multi-mediating relationship between the social value, hedonic value, and novelty value of perceived products was proposed. The study's results address the need for more consumer research in the intelligent seating industry, as well as how to give researchers and marketing firms solutions and suggestions based on facts.

**Keywords:** smart safety seat; domain-specific innovation; consumption value; partial least squares structural equation model (PLS-SEM); multiple parallel mediations

## 1. Introduction

China now has the most car ownership in the world due to its rapid economic development. As of the end of November 2022, there were 415 million motor vehicles in China overall, of which 318 million were automobiles, according to the most recent statistics provided by the Ministry of Public Security of China. More than 500 million people drive automobiles, including 463 million car owners [1]. The number of cars and drivers in China currently leads the world. However, child safety seats are not used nearly enough in China. According to the Blue Book of China Children's Road Traffic Safety (2018) [2], the death rate of children traveling in vehicles without child safety seats is eight times greater than that of children traveling in vehicles with child safety seats installed. Less than 10% of children in China use child safety seats, much lower than the nearly 90% penetration rate in Europe and the US. Sixty percent of traffic fatalities among children aged 1 to 14 occurred in the ten years from 2012 to 2021 among children under six. Such information was made public by Gao Yan, director of the Traffic Safety Technology Department of the Traffic Management Science Institute of the Ministry of Public Security [3]. Children who died in cars made up a more significant percentage of child traffic injury deaths, accounting for an average of 43.9% over the past ten years. An amount of 2954 kids between the ages of 1 and 15 lost their lives in traffic accidents, and 13,938 others were hurt, according to the China Road Traffic Accident Statistics Annual Report (2017) [4]. It is critical to increase the use of child safety seats as a result.

There have been efforts made in every sphere of society to increase the use of child safety seats. For instance, some researchers have studied the data on child safety seats manufactured and exported from China [5]. In contrast, some designers have conducted research to enhance the functionality of child safety seats [6–8]. Some doctors have also studied the situation of newborn children using child safety seats [9–11]. Studies have also urged the government and law enforcement agencies to make laws and regulations more stringent [12–14]. The analysis of the deciding factors in purchasing child safety seats needs to be improved from the consumer's perspective.

The theory of consumption value (TCV) is a marketing theory that provides insights into the motivations of consumers' consumption behavior through consumption value [15]. Customers prefer to purchase goods that are perceived to have the highest value [16]. The theory of consumer value (TCV), developed by Sheth et al., provides a substantial theoretical and practical contribution by illuminating the rationale behind consumers' decisions to purchase specific products, product categories, and brands based on their perceived value [17]. TCV offers the chance to enlighten and enrich the knowledge of how a variety of goods and services are consumed, including food [18,19], clothing [20], health [21], education [22], travel [23], etc. This claim is supported by recent research, which shows that, in order to comprehend consumer behavior in the modern online and offline environments, perceived value tends to present consumption value [24,25]. TCV has made significant contributions to the theory and practice of motivating consumer behavior. There are no studies on the connection between consumer value and consumer behavior in the context of intelligent safety seats. This study's goals are to integrate and synthesize the knowledge based on TCV's research findings in the intelligent products field, conduct a more profound analysis of smart child safety seats, and make recommendations for future consumer studies.

China leads the world in producing child safety seats, turning out more than 15 million units annually. To meet consumer demand and increase the use of child safety seats by aligning with consumer values. Chinese manufacturers have developed a variety of "innovative" and functional product designs: the intelligent safety seat. Dehumidification and safety monitoring systems integrated into innovative safety seats are examples [26,27]. In addition, child safety seats that can adjust to a child's size automatically [28] and detect and alarm when a child has been left in a vehicle [29] are now commercially available. At the moment, there are no research studies on intelligent child safety seats, which are an innovative intelligent product in a particular field.

The adoption of new products by consumers has been extensively studied in the past, and several conventional research models have been put forth. For instance, TAM (the technology adoption model) [30], TAM2 (the enhanced technology acceptance model) [31], TPB (the planned behavior theory) [32], TRA (the rational action theory) [33], UTAUT (the unified theory of technology acceptance and use [34], and DSI (the domain-specific innovativeness) have been developed [35].

TAM, TAM2, TRA, TPB, and UTAUT typically study new science and technology. Researchers are more likely to use the DSI theory for research on innovative products in various fields and attempt to explain its utility to consumer research [36,37].

DSI has been found to be the most practical scale for assessing consumers' capacity for innovation in particular product categories in empirical studies conducted across the globe, including those conducted in the United States, Germany, and France [37–39].

The DSI structure has been found to positively influence consumers' willingness to try new products in earlier studies with related properties, though the influence coefficient is very small [38,39]. Because of this, some researchers have reformulated and put to the test the DSI structure, as well as its relationship to the characteristics of innovative products, and have investigated how these attributes affect consumers' intentions to buy new products [40,41].

Since there is no similar research in the intelligent child safety seat industry, we refer to Jeong, S. C. et al. [40,41]. We modified the research model to be more suitable for consumers of intelligent child safety seats.

Prior academic work has yet to, to our knowledge, successfully integrate domain-specific innovation theories into the theoretical structure of consumer value. This study aims to fill knowledge gaps by examining the relationship between innovation indicators in these areas and how consumers perceive the value of smart safety seats and whether they plan to purchase them.

The contributions of this study are as follows:

(1) Extend the application of partial least squares structural equation modeling (PLS-SEM) to a new subject area.
(2) Broaden the research perspective of consumer motivation.
(3) The user portrait of new product promotion is analyzed from two perspectives: consumer product innovation and consumer information innovation.
(4) People now have a new perspective on consumer behavior as a result of the integration of the theory of consumer value and the theory of innovation in particular fields.

This paper's remaining sections are organized as follows. The first section introduces the theoretical foundation of this study. The second section discusses the study's methodology, followed by the third section's study results. The fourth section provides suggestions for future research directions. The fifth section introduces the study's limitations and conclusions.

## 2. Theoretical Background and Hypothesis Development

### 2.1. Domain-Specific Innovation (DSI)

According to pertinent marketing research, consumer innovativeness, or the ability to use a new product more quickly than other consumers, is one of the most significant personality attributes influencing a consumer's decision to use or accept a new product [42,43]. According to the DSI structure, individuals who are consumer innovative in one category do not necessarily exhibit this behavior in other fields, which concentrates on aspects of human behavior related to innovation within a person's particular sphere of interest [44]. Thus, defined in the field of innovation performance more precisely forecasts the adoption of new products, leading to the development of innovation (domain-specific innovativeness) in specific areas, using innovative specific areas to forecast consumers' specific interest in the field of new products early adoption behavior and attitude [45]. Researchers worldwide have been adapting this model's structure to new fields and products in recent years [46–51]. However, many studies look at the product side of things, concentrating on the early degree of customers' adoption of new items while disregarding that some people only pay attention to the information about new products but do not necessarily buy them. Companies today need to pay closer attention to innovation centered on product details because of the dynamic nature of the business and technological landscapes and individual consumers' penchant for constant reinvention. Based on the two pillars of product and information, customer innovation in a given industry can be broken down into two categories: consumer information innovation and consumer product innovation [40].

### 2.2. Theory of Consumption Value (TCV)

TCV is a multifaceted method that evaluates consumption value from a behavioral standpoint and offers different kinds of perceived value [52]. TCV was first presented by Sheth et al. [17] in the Journal of Business Research article "Why We Buy What We Buy: A Theory of Consumer Value.". By emphasizing the value of consumption to anticipate, characterize, and justify decision behavior, this theory illuminates the driving force behind consumer behavior. TCV offers a multidisciplinary perspective for examining consumer choice behavior since Sheth et al. [17] employed a wide range of disciplines, including economics, marketing, consumer behavior, sociology, and psychology, to establish their

theories and their values. The creators of the theory emphasize that only individual, deliberate, and free decision-making is applicable in practice.

When evaluating a marketing offer's differences from competing offerings, consumers must consider all of the benefits and expenses involved. This is known as perceived value [16]. The complexity of perceived worth has been proposed by numerous scientists [53–56]. As a result, they suggest various dimensions to look at perceived value. For instance, the proposal of Chahal, Hardeep, Kumari, and Neetu (2012) for perceived value has a multidimensional structural character that is produced by 27 items spread over six dimensions that are significant for consumer value measurement. The dimensions include acquisition value, transaction value, efficiency value, aesthetic value, social interaction value, and self-gratification value [54].

According to Sweeney and Soutar (2001), the PERVAL scale—which is based on the utilitarian and hedonistic construction of this tendency—proposes emotional, social, quality, and price values as dimensions of consumption value [53]. Later, El-Adly, Mohammed, and Ismail (2019) offered a different typology for customer value that combined self-gratification value, aesthetic value, price value, prestige value, transaction value, hedonic value, and quality value [56].

More specifically, Sheth et al. [24] are devoted to answering the following questions: "Why do consumers choose to buy or not buy (or use or not use) particular products? Why do consumers choose one type over another, and why do consumers choose one brand over another?". In order to answer these questions in TCV, we propose four kinds of consumer perceived value: perceived usefulness value (PUV), perceived social value (PSV), perceived hedonic value (PHV), and perceived novelty value (PNV).

*2.3. Hypothesis Development*

We combine a modified version of the DSI theory with the TCV theory to examine the behavioral intention of consumers to purchase innovative car safety seats. The model says that consumer information innovation, consumer product innovation, and perceived value will significantly impact people's desire to buy innovative safety seats. As a mediating variable, perceived value affects the intended effect of consumer innovation in a specific field on consumers' purchase of innovative car safety seats. Figure 1 shows the model without mediating variables, and Figure 2 shows the model with mediating variables.

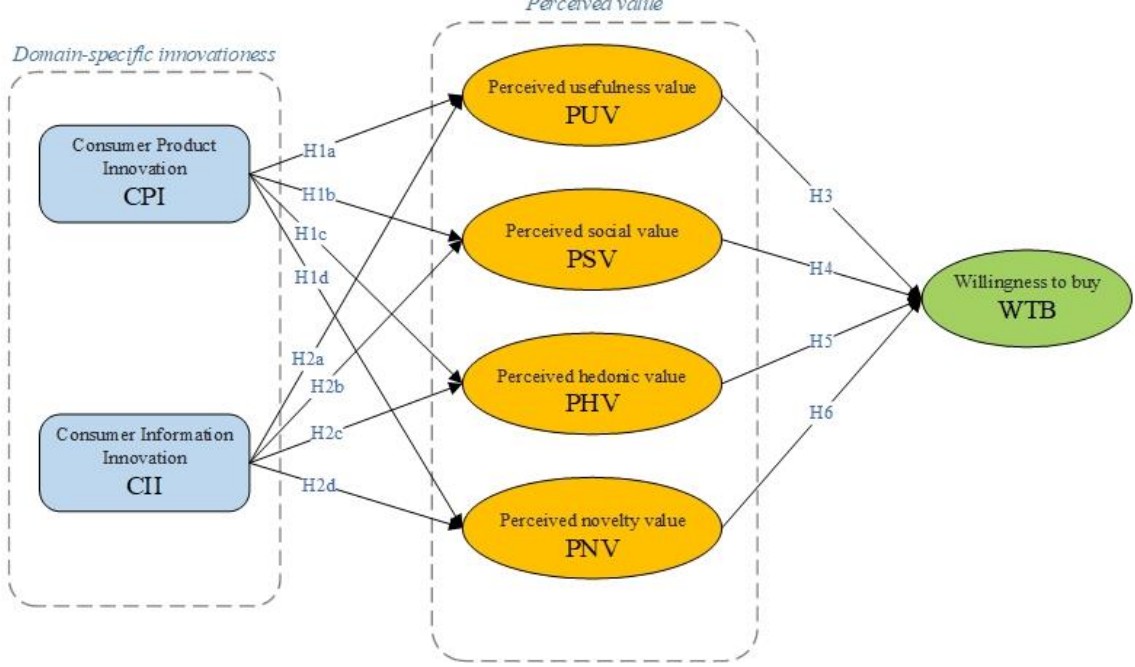

**Figure 1.** Research model A. (without mediator).

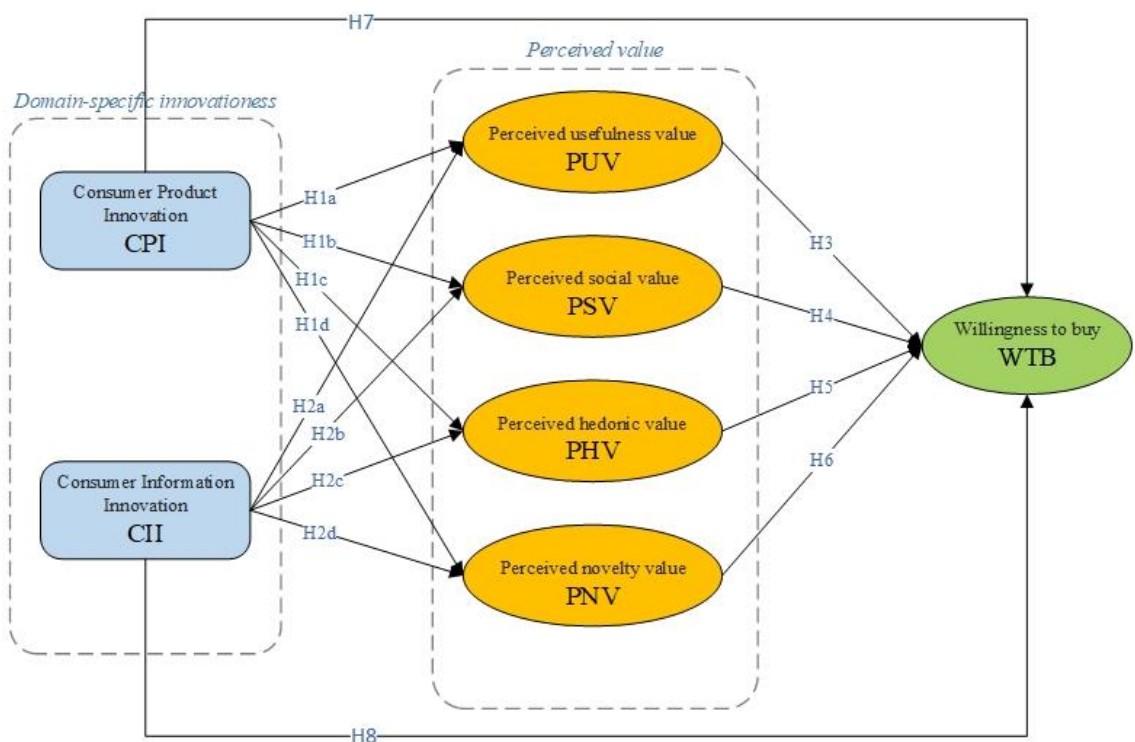

**Figure 2.** Research model B. (Perceived value as a mediator for CPI and WTB; Perceived value as a mediator for CII and WTB).

### 2.3.1. Consumer Product Innovation and Perceived Value

Consumer product innovation (CPI) measures how quickly a product's user base adopts new features and technologies relative to other products in the same market segment [57].

According to previous studies [58], consumers on the cutting edge of innovation tend to be curious. Customers such as them are receptive to new concepts and eagerly anticipate the arrival of innovative products. As a result, people with a high CPI may be more aware of the worth of new things, more possessive, and more prone to believe that material goods may bring them happiness [59]. Studies have shown that product innovation can increase the perceived value of customers [60].

**H1a.** *Consumer product innovation (CPI) has a positive effect on perceived usefulness value (PUV).*

**H1b.** *Consumer product innovation (CPI) has a positive effect on perceived social value (PSV).*

**H1c.** *Consumer product innovation (CPI) has a positive effect on perceived hedonic value (PHV).*

**H1d.** *Consumer product innovation (CPI) has a positive effect on perceived novelty value (PNV).*

### 2.3.2. Consumer Information Innovation and Perceived Value

Consumer information innovation (CII) refers to the propensity for consumers to be exposed to and receive information about new items before others in a specific product category, but not necessarily to acquire the new product. Expertly informed innovative consumers are eager to learn about innovative new products and desire access to them even if they do not possess them [61].

Previous research has studied a similar issue [62], defining and conceptualizing innovativeness as a person's desire to learn about a new product rather than purchase the thing. Moreover, Hartman et al. [63] consider this form of innovation one of the three unique dimensions of innovation. It shows that customers with a high CII score may hunt for information outside their personal knowledge base to review their memory and knowledge about the product or service gleaned from past experiences to make purchasing

decisions. They enjoy sharing more knowledge [64]. External search enables customers to acquire and preserve adequate information for decision-making [24]. Al-Rahmi, W. et al. 's investigation of students' intention to use e-learning systems found that information innovation can affect students' value-oriented perception [65].

**H2a.** *Consumer information innovation (CII) has a positive effect on perceived usefulness value (PUV).*

**H2b.** *Consumer information innovation (CII) has a positive effect on perceived social value (PSV).*

**H2c.** *Consumer information innovation (CII) has a positive effect on perceived hedonic value (PHV).*

**H2d.** *Consumer information innovation (CII) has a positive effect on perceived novelty value (PNV).*

### 2.3.3. Perceived Usefulness Value and Willingness to Buy

There is a link between consumer behavior and individual value, according to the theory of consumption value [24]. It is asserted that customers evaluate a product's relevance and desirability based on how those features relate to the individual consequences of product use. Similarly to this, the applicability and desirableness of personal consequences are determined by their relationship to a consumer's values.

The usefulness value of products is defined as "the extent to which a person believes that the use of a certain system improves his/her work performance". The previous literature showed that college students perceived the usefulness of educational social network sites had a moderate impact on their willingness to use educational social network sites [66]. Studies have shown that consumers accept that the behavior intention of mobile banking in Malaysia is affected by perceived usefulness [67].

**H3.** *Perceived usefulness value (PUV) has a positive effect on users' purchase intention (WTB).*

### 2.3.4. Perceived Social Value and Willingness to Buy

Perceived value is essential to consumer satisfaction and behavior [68]. Numerous empirical studies have demonstrated the beneficial relationship between perceived value and customer action intentions [55,69–71]. In line with these findings, the perceived value of an innovative car safety seat may drive consumers to acquire this product.

Previous studies have found that social value may directly affect consumers' purchase intention for green products [72]. Jaleel, A. et al. evaluated the relationship between consumers' perceived value and behavioral intention in medical tourism services and found that perceived social influence and social value had a great impact on usage intention [73].

**H4.** *Perceived social value (PSV) has a positive effect on users' purchase intention (WTB).*

### 2.3.5. Perceived Hedonic Value and Willingness to Buy

Hedonic value refers to the feeling of pleasure, comfort, safety, comfort, and relaxation generated when using a particular brand. The inherent hedonic value of a brand can create brand personality attraction, which can influence consumers' perception of the attributes of products or services and meet their expectations. Hedonic value positively impacts consumers' behavioral intentions [74].

**H5.** *Perceived hedonic value (PHV) has a positive effect on users' purchase intention (WTB).*

### 2.3.6. Perceived Novelty Value and Willingness to Buy

Fazal-e-Hasan, S. et al. found that novelty can significantly improve consumers' intention to use intelligent retail technology [75]. Some scholars have found that customers want to experience new services, and the functions of these services should differ from the original ones. If customers' expectations of exploration and learning are met, consumers' usage intentions will increase [76]. A recent study by Adapa. S. et al. established a positive correlation between perceived novelty and use intent [77]. Therefore, we assume that:

**H6.** *Perceived novelty value (PNV) has a positive effect on users' purchase intention (WTB).*

2.3.7. Innovation in Specific Areas and Willingness to Buy

An example of DSI is consumers' propensity to be the first adopters of new offerings within a certain market segment [78]. While studies in various disciplines have looked into why and how consumers try new products [41,51,79,80], the bulk of the literature has focused on what motivates consumers to try new things in the first place. The idea of domain-specific consumer innovativeness is commonly utilized in diffusion theory to explain why and how people in a given social system accept and spread new products [81]. Evidence shows that individuals' propensity for innovation influences how quickly they adopt new technologies [82].

In this case, consumers in specific areas that may have certain products or services tend to show greater purchase willingness than those without such products or services. In other words, innovative consumers tend to have higher consumption propensity than conservative consumers [42]. Therefore, suppose:

**H7.** *Consumer product innovation (CPI) has a positive effect on users' purchase intention (WTB).*

**H8.** *Consumer information innovation (CII) has a positive effect on users' purchase intention (WTB).*

2.3.8. The Mediating Role of the Perceived Value

In general, innovators or early adopters of innovative products, who have a positive attitude towards innovation, assess value differently than most late consumers or laggards. The innate innovation ability of consumers stimulates innovators to perceive the added value of innovative products. High and positive perceived value drives innovators to adopt new products [83]. Perceived value plays a mediating role in the influence of social identity and social influence on the purchase intention of new products. Al-Jundi, S.A. et al. found that domain-specific innovation mediates between consumers' innate innovation and their intention to buy new products [41]. At the same time, Hong et al. proved that hedonic and utilitarian values mediate the influence of smartwatch consumers' innovation on their intention to continue [84]. Therefore, the following hypothesis is proposed:

**H7a.** Perceived usefulness value (PUV) mediates the effect of consumer product innovativeness (CPI) on users' purchase intention (WTB).

**H7b.** *Perceived social value (PSV) mediates the effect of consumer product innovativeness (CPI) on users' purchase intention (WTB).*

**H7c.** *Perceived hedonic value (PHV) mediates the effect of consumer product innovativeness (CPI) on users' purchase intention (WTB).*

**H7d.** *Perceived novelty value (PNV) mediates the effect of consumer product innovativeness (CPI) on users' purchase intention (WTB).*

**H8a.** *Perceived usefulness value (PUV) mediates the effect of consumer information innovativeness (CPI) on users' purchase intention (WTB).*

**H8b.** *Perceived social value (PSV) mediates the effect of consumer information innovativeness (CPI) on users' purchase intention (WTB).*

**H8c.** *Perceived hedonic value (PHV) mediates the effect of consumer information innovativeness (CPI) on users' purchase intention (WTB).*

**H8d.** *Perceived novelty value (PNV) mediates the effect of consumer information innovativeness (CPI) on users' purchase intention (WTB).*

## 3. Materials and Methods

### 3.1. Participants and Procedures

Through professional research organizations, a questionnaire survey is administered to clients of mother and child stores and the automobile sales service shop for Bentley in major developed cities (Shanghai, Beijing, and Hangzhou) in China as part of this study. The questionnaire for this study is posted on wenquan.com (China's largest website for questionnaire surveys), and a total of 1200 respondents were questioned utilizing a combination of convenience sampling and purposive sampling. The purpose of purposive sampling is to find out the parents with children and private cars and survey them. All of the respondents were parents of small children.

The research goals were established by carefully selecting samples from the smart child safety seats currently available on the Chinese market. Child safety seats with various smart technologies were ranked. The ForU smart child safety seat was chosen as a target to avoid the influence of individual preferences for specific features on willingness to pay. The intelligent child safety seat designed by the Shanghai Woyoo Electronic Technology Co., Ltd. was sold in 2018. ForU's smart child safety seat is the case for this study for the following reasons. First of all, the ForU intelligent child safety seat, which is currently sold in China, has a lot of functions and patents [85–106]. Secondly, the ForU smart child safety seat uses sensors to detect when a child is in the seat, alarms when the temperature rises, and turns on ventilation to help the child cool down. If the driver forgets the child in the vehicle, the mobile phone will notify and alarm the driver. The automated ISOFIX locking, heated, ventilated, and foldable smart child safety seat are all available. Additionally, the intelligent child safety seat has an APP that lets users control the backrest's angle and the heating and ventilation systems with their voice and phone. Thirdly, customers can easily purchase this sophisticated child safety seat from Bentley and Taobao stores' automobile sales service shops.

### 3.2. Questionnaire Design

The questionnaire consists of four parts. The first part is about the introduction of innovative child safety seats. The second part is the population and tourism information questionnaire, which investigated the demographic data, annual family income, and family travel frequency. The third part is the DSI scale, which is used to measure specific categories of consumer innovation [63,80,107]. The fourth part is the questionnaire TCV model. The scale was created by the TC.

The TCV model includes four subscales: perceived usefulness value, perceived social value, perceived hedonic value, and perceived novelty value [34,108–111]. All scales are 7-point Likert scales with 28 questions. According to the suggestion of bilingual professor Zhao Mei [112], the scale was first translated into Chinese according to the language expression, cultural background, and customs, and then it was translated back to English for proofreading with the original scale. Table A1 shows the questions for the questionnaire in Appendix A.

### 3.3. Data Collection

We conducted on-site trials of smart safety seats in shopping malls and auto 4S stores to understand consumers' feelings better. Consumers were required to fill out questionnaires after the trials. We have included a detailed introduction to the smart safety seat in the first section of the questionnaire for users who cannot enter the place to try out the smart safety seat. Consumers can clearly understand the smart safety seat's function, appearance, and characteristics thanks to these introductions. Respondents were required to complete the questionnaire.

Respondents were asked to complete the questionnaire online through Wechat or a browser. Professional research institutions that set the same IP address cannot submit the questionnaire repeatedly, and the same Wechat can only participate in the questionnaire once to improve the validity of the data. Wechat accounts and suspicious IP addresses have

been blocked. All respondents were informed that the information was private and would not be publicized. They voluntarily completed the questionnaire and were rewarded with 2 yuan (CNY) in exchange. We collected 1152 questionnaires between March and July 2022. We rescreened the data, removing questionnaires with the same score for all options, and obtained 1057 valid responses.

Table 1 provides an overview of the respondents' age, gender, educational background, household income, and frequency of trips with children. Of the respondents, 38.7% (409) were female, and 61.3% (648) were male, both of whom were parents of existing children. The majority, about 77.6%, were reported to be between the ages of 26 and 45. More than 90% had higher education. The largest proportion of households with an annual household income of RMB 100,000–300,000 yuan was 46.2%.

**Table 1.** Demographic profile of respondents (N = 1057).

| Demographic Profile | Frequency | Relative Frequency (%) |
|---|---|---|
| Gender | | |
| Female | 409 | 38.7 |
| Male | 648 | 61.3 |
| Age (years) | | |
| $\leq 25$ | 198 | 18.7 |
| 26–35 | 598 | 56.6 |
| 36–45 | 222 | 21.0 |
| >45 | 39 | 3.7 |
| Education level | | |
| 10–12 years | 98 | 9.3 |
| College degree | 190 | 18.0 |
| Bachelor | 664 | 62.8 |
| Post-graduate degree/PhD | 105 | 9.9 |
| Annual household income (RMB) | | |
| $\leq 100,000$ Yuan | 185 | 17.5 |
| 100,000–300,000 Yuan | 488 | 46.2 |
| 300,000–500,000 Yuan | 248 | 23.5 |
| >500,000 Yuan | 136 | 12.9 |
| Travel times (per month) | | |
| 0–10 times | 662 | 62.6 |
| 11–30 times | 395 | 27.4 |

*3.4. Data Analysis*

Using Smart PLS 2.0, partial least squares structural equation modeling was performed. Smart PLS's strengths lie in its flexibility [113]; it can be applied to the analysis of non-normal data or studies with small sample sizes, as well as to the analysis of more sophisticated multi-order latent variable models and the exploration of novel models [114,115]. The model is complicated since it is a second-order model with seven latent variables, and the data used in this study are not strictly normally distributed. This exploratory model has not been the subject of any prior research. The aforementioned considerations led to the choice of the partial least squares structural equation modeling method for this investigation.

The useful data were examined in three stages; the first two steps are analyzed according to the two-step method proposed by Anderson and Gerbing [116].

Step 1: Descriptive statistics were run on the population data. Regarding the scale, the correlation between latent variables and observable variables was used to assess measurement models. Reliability tests and validity tests were run on the data. The validity tests were further broken down into convergent and discriminant validity tests.

Step 2: Examine the structural equation model, which comprises the paths that latent variables take to interact with each other. Pay special attention to the regression weight

and significance level, as well as the amount of variance that these latent variables explain. Validity assessment of structural models using the blindfolding procedure was conducted.

Step 3: Out-of-sample prediction [117,118].

## 4. Results

### 4.1. Measurement Model

Table 2 is the descriptive statistics of respondents' answers to the Likert scale.

**Table 2.** Descriptive statistics.

| | N | Min. | Max. | M | SD | Relative Frequencies (%) | | | | | | |
|---|---|---|---|---|---|---|---|---|---|---|---|---|
| | | | | | | 1 | 2 | 3 | 4 | 5 | 6 | 7 |
| CPI1 | 1057 | 1 | 7 | 4.77 | 1.17 | 0.76 | 1.42 | 8.33 | 32.54 | 33.21 | 14.19 | 9.56 |
| CPI2 | 1057 | 1 | 7 | 4.53 | 1.19 | 0.57 | 2.93 | 15.42 | 29.90 | 33.11 | 11.73 | 6.34 |
| CPI3 | 1057 | 1 | 7 | 4.68 | 1.15 | 0.38 | 1.42 | 10.88 | 34.44 | 31.79 | 12.58 | 8.51 |
| CII1 | 1057 | 1 | 7 | 4.47 | 1.24 | 0.95 | 2.65 | 17.98 | 30.94 | 29.52 | 10.50 | 7.47 |
| CII2 | 1057 | 1 | 7 | 4.51 | 1.27 | 1.42 | 3.50 | 14.76 | 30.75 | 29.61 | 12.68 | 7.28 |
| CII3 | 1057 | 1 | 7 | 4.53 | 1.26 | 1.42 | 2.93 | 14.95 | 29.80 | 31.69 | 11.54 | 7.66 |
| PUV1 | 1057 | 1 | 7 | 4.89 | 1.14 | 0.47 | 0.95 | 6.72 | 29.14 | 37.09 | 14.66 | 10.97 |
| PUV2 | 1057 | 1 | 7 | 4.85 | 1.18 | 0.57 | 0.95 | 9.46 | 25.92 | 39.55 | 11.83 | 11.73 |
| PUV3 | 1057 | 1 | 7 | 4.85 | 1.17 | 0.47 | 1.04 | 9.37 | 26.96 | 37.28 | 13.91 | 10.97 |
| PUV4 | 1057 | 1 | 7 | 4.72 | 1.18 | 0.76 | 1.42 | 10.12 | 31.88 | 33.68 | 13.06 | 9.08 |
| PSV1 | 1057 | 1 | 7 | 4.73 | 1.20 | 0.66 | 1.61 | 10.12 | 32.45 | 32.17 | 12.58 | 10.41 |
| PSV2 | 1057 | 1 | 7 | 4.65 | 1.23 | 1.32 | 1.80 | 12.30 | 30.37 | 32.36 | 13.53 | 8.33 |
| PSV3 | 1057 | 1 | 7 | 4.57 | 1.25 | 1.42 | 1.99 | 14.29 | 31.41 | 29.90 | 13.06 | 7.95 |
| PHV1 | 1057 | 1 | 7 | 4.61 | 1.24 | 1.04 | 2.18 | 13.62 | 30.56 | 30.84 | 13.53 | 8.23 |
| PHV2 | 1057 | 1 | 7 | 4.79 | 1.21 | 0.57 | 1.80 | 10.03 | 29.04 | 33.87 | 13.91 | 10.79 |
| PHV3 | 1057 | 1 | 7 | 4.73 | 1.20 | 0.76 | 1.42 | 10.50 | 31.22 | 33.11 | 13.25 | 9.74 |
| PNV1 | 1057 | 1 | 7 | 4.77 | 1.20 | 0.38 | 1.80 | 10.69 | 28.48 | 35.38 | 12.49 | 10.79 |
| PNV2 | 1057 | 1 | 7 | 4.44 | 1.24 | 1.32 | 2.93 | 16.93 | 31.69 | 29.52 | 11.26 | 6.34 |
| PNV3 | 1057 | 1 | 7 | 4.71 | 1.18 | 0.57 | 2.18 | 10.41 | 29.80 | 34.63 | 14.19 | 8.23 |
| WTB1 | 1057 | 1 | 7 | 4.85 | 1.19 | 0.57 | 0.95 | 8.99 | 28.67 | 36.23 | 12.39 | 12.20 |
| WTB2 | 1057 | 1 | 7 | 4.79 | 1.23 | 0.66 | 1.89 | 9.65 | 28.86 | 35.19 | 11.83 | 11.92 |
| WTB3 | 1057 | 1 | 7 | 4.70 | 1.18 | 0.76 | 1.51 | 10.22 | 32.07 | 34.63 | 11.64 | 9.18 |

Note: N = number; Min. = Minimum value; Max. = Maximum value; M = Mean value; SD = standard deviations; CII = consumer information innovation; CPI = consumer product innovation; PHV = perceived hedonic value; PNV = perceived novelty value; PSV = perceived social value; PUV = perceived usefulness value; WTB = willing to buy; 1 = completely disagree; 2 = disagree; 3 = partially disagree; 4 = neutral; 5 = partially agree; 6 = agree; 7 = completely agree.

Table 3 displays the findings of the investigation of reliability. It can be seen that Cronbach's $\alpha$ of the seven latent variables is between 0.841 and 0.896, which are all higher than the benchmark value of 0.70. The composite reliability values are between 0.904 and 0.933, which were higher than 0.70, indicating that the reliability of the questionnaire is good, and the collected data are reliable. All item factor loadings should be more than 0.70, as per the Fornell and Larcker criteria [119], and the AVE for each construct was greater than 0.50 [120]. As can be seen in Table 3, the data for this model all met the requirements, indicating strong convergent validity.

Discriminant validity was tested using the Fornell-Larcker criterion [119], which is a measure of the expected level of "difference" between items for different factors. To test the discriminant validity of the model, the AVE of each factor was compared with the correlation square. The value on the diagonal is the square root of the AVE, and the other values are the correlation coefficients between the factors, which are considered to have good discriminant validity when the AVE is greater than the correlation coefficient between the factor and the other factors. From Tables 4 and 5, it can be seen that the model has good discriminant validity.

**Table 3.** Reliability and convergent validity test of the measurement model.

| Construct | Items | Loadings | Cronbach | CR | AVE | $R^2$ |
|---|---|---|---|---|---|---|
| Consumer information innovation (CII) | You enjoy learning about new information technology. | 0.910 | 0.892 | 0.933 | 0.822 | |
| | Compared to your contemporaries, you are often more sensitive to knowledge about novel things. | 0.900 | | | | |
| | You tend to be more interested than your peers in the capabilities and applications of new information technology. | 0.910 | | | | |
| Consumer product innovation (CPI) | Compared to your peers, you typically own more smart products. | 0.913 | 0.883 | 0.927 | 0.81 | |
| | You frequently purchase new smart devices before your peers. | 0.876 | | | | |
| | You typically opt to purchase the newest smart devices. | 0.911 | | | | |
| Perceived hedonic value (PHV) | Using the smart child safety seat would be fun during car trips. | 0.899 | 0.877 | 0.924 | 0.803 | 0.637 |
| | Using the smart child safety seat would be enjoyable. | 0.888 | | | | |
| | Using the smart child safety seat would make me and my kids very happy during car trips. | 0.901 | | | | |
| Perceived novelty value (PNV) | The Smart child safety seat is a new and refreshing device. | 0.893 | 0.841 | 0.904 | 0.759 | 0.641 |
| | Smart child safety seats are unique in this. | 0.829 | | | | |
| | I think using a smart child safety seat is a novel experience. | 0.891 | | | | |
| Perceived social value (PSV) | I should use smart child safety seats, according to people that matter to me. | 0.880 | 0.86 | 0.914 | 0.781 | 0.626 |
| | Those who have the power to affect my behavior believe I should employ smart child safety seats. | 0.878 | | | | |
| | People whose viewpoints I respect want me to use smart child safety seats. | 0.892 | | | | |
| Perceived usefulness value (PUV) | I think the smart child safety seat would be good for my driving. | 0.905 | 0.896 | 0.928 | 0.763 | 0.643 |
| | I could go to my destination more safely if I used the smart child safety seat. | 0.859 | | | | |
| | The smart child safety seat would make it easier for me to go where I am. | 0.879 | | | | |
| | I could reach my destination faster if I used the smart child safety seat. | 0.851 | | | | |
| Willing to buy (WTB) | If I can afford it, I would prefer to buy a smart child safety seat | 0.901 | 0.872 | 0.921 | 0.796 | 0.797 |
| | I intend to use smart child safety seats in the future. | 0.885 | | | | |
| | I would want to try the smart child safety seat. | 0.891 | | | | |

Note: loadings = factor loading; Cronbach = Cronbach's alpha; CR = construct reliability; AVE = average variance extracted, a measure of convergence among observable variables reflecting a latent variable; R2 = coefficient of determination [121]. CII = consumer information innovation; CPI = consumer product innovation; PHV = perceived hedonic value; PNV = perceived novelty value; PSV = perceived social value; PUV = perceived usefulness value; WTB = willing to buy.

*4.2. Structural Model*

Evaluate the three research models proposed by the authors according to the research of Hair et al. [115,122]. The hypotheses are tested under these three different conceptual models, and the influences of independent and mediating variables on dependent variables are analyzed.

**Table 4.** Cross-loadings values for each block of indicators.

|      | CII   | CPI   | PIV   | PNV   | PSV   | PUV   | WTB   |
|------|-------|-------|-------|-------|-------|-------|-------|
| CII1 | 0.910 | 0.753 | 0.662 | 0.665 | 0.686 | 0.626 | 0.650 |
| CII2 | 0.900 | 0.781 | 0.658 | 0.667 | 0.662 | 0.636 | 0.654 |
| CII3 | 0.910 | 0.757 | 0.669 | 0.653 | 0.654 | 0.646 | 0.658 |
| CPI1 | 0.736 | 0.913 | 0.722 | 0.727 | 0.696 | 0.748 | 0.739 |
| CPI2 | 0.789 | 0.876 | 0.707 | 0.698 | 0.699 | 0.689 | 0.711 |
| CPI3 | 0.750 | 0.911 | 0.700 | 0.714 | 0.703 | 0.722 | 0.717 |
| PHV1 | 0.679 | 0.715 | 0.899 | 0.772 | 0.762 | 0.741 | 0.740 |
| PHV2 | 0.641 | 0.706 | 0.888 | 0.783 | 0.736 | 0.775 | 0.751 |
| PHV3 | 0.645 | 0.699 | 0.901 | 0.755 | 0.746 | 0.761 | 0.775 |
| PNV1 | 0.631 | 0.718 | 0.771 | 0.893 | 0.728 | 0.772 | 0.772 |
| PNV2 | 0.648 | 0.646 | 0.712 | 0.829 | 0.690 | 0.661 | 0.670 |
| PNV3 | 0.632 | 0.705 | 0.762 | 0.891 | 0.713 | 0.752 | 0.748 |
| PSV1 | 0.632 | 0.687 | 0.741 | 0.712 | 0.880 | 0.755 | 0.739 |
| PSV2 | 0.651 | 0.692 | 0.736 | 0.724 | 0.878 | 0.729 | 0.707 |
| PSV3 | 0.668 | 0.680 | 0.735 | 0.724 | 0.892 | 0.708 | 0.701 |
| PUV1 | 0.623 | 0.736 | 0.747 | 0.746 | 0.728 | 0.905 | 0.767 |
| PUV2 | 0.591 | 0.686 | 0.727 | 0.719 | 0.701 | 0.859 | 0.742 |
| PUV3 | 0.625 | 0.716 | 0.770 | 0.752 | 0.749 | 0.879 | 0.744 |
| PUV4 | 0.613 | 0.655 | 0.716 | 0.710 | 0.713 | 0.851 | 0.709 |
| WTB1 | 0.629 | 0.721 | 0.772 | 0.737 | 0.726 | 0.784 | 0.901 |
| WTB2 | 0.640 | 0.713 | 0.740 | 0.775 | 0.720 | 0.749 | 0.885 |
| WTB3 | 0.662 | 0.714 | 0.746 | 0.735 | 0.722 | 0.735 | 0.891 |

**Table 5.** Discriminant validity matrix (Fornell and Larcker criterion).

|      | CII    | CPI    | PIV    | PNV    | PSV    | PUV    | WTB    |
|------|--------|--------|--------|--------|--------|--------|--------|
| CII  | **0.907** |        |        |        |        |        |        |
| CPI  | 0.842  | **0.900** |        |        |        |        |        |
| PHV  | 0.731  | 0.789  | **0.896** |        |        |        |        |
| PNV  | 0.730  | 0.792  | 0.859  | **0.871** |        |        |        |
| PSV  | 0.736  | 0.777  | 0.835  | 0.815  | **0.884** |        |        |
| PUV  | 0.702  | 0.800  | 0.847  | 0.838  | 0.827  | **0.874** |        |
| WTB  | 0.721  | 0.803  | 0.843  | 0.839  | 0.810  | 0.848  | **0.892** |

Note: Values (bold) on the diagonal represent the square root of the AVE, while the off-diagonals are correlations.

4.2.1. Research Model A. (without Mediator)

Each path coefficient's statistical significance was evaluated using t-tests, and as noted before, bootstrapping (5000 sub-samples) was utilized to do so.

Table 6 shows the results of direct effects analysis show that the direct effects are significant. The results of the indirect effect analysis show that the indirect effect is significant.

As seen in Figure 3 and Table 6, CPI had a significant positive effect on PUV (t = 17.950, $p < 0.001$), PSV (t = 12.116, $p < 0.001$), PHV (t = 13.867, $p < 0.001$), and PNV (t = 1.690, $p < 0.001$). CII had a significant positive effect on PUV (t = 2.159, $p < 0.05$), PSV (t = 6.065, $p < 0.001$), PHV (t = 4.765, $p < 0.001$), and PNV (t = 4.655, $p < 0.001$) also had a significant positive effect, but it was significantly weaker than the CPI effect on them. PUV (t = 6.537, $p < 0.001$), PSV (t = 3.330, $p < 0.01$), PHV (t = 4.673, $p < 0.001$), and PNV (t = 5.938, $p < 0.001$) all had significant positive effects on WTB.

The value of the coefficient of determination $R^2$, which also explains the variance of the regression model, is in the range of [0, 1], and the closer it is to 1, the more the independent variable can explain the variance of the dependent variable, and the larger the value, the better [123]. According to Hair, "0.25 is weak, 0.50 is moderate, and 0.75 is the model's substantial explanatory power" [115]. Table 6 shows that PUV, PSV, PHV, and PNV are moderate, and WTB is strong.

**Table 6.** Hypothesized relationships for all effects.

| Hypotheses | Path | Estimate | Standard Error | T-Value | *p* Value | 95%CI | $f^2$ | Supported |
|---|---|---|---|---|---|---|---|---|
| | Direct effects | | | | | | | |
| H2c | CII→PHV | 0.232 *** | 0.049 | 4.765 | 0.000 | [0.132; 0.324] | 0.043 [a] | Yes |
| H2d | CII→PNV | 0.215 *** | 0.046 | 4.655 | 0.000 | [0.122, 0.303] | 0.037 [a] | Yes |
| H2b | CII→PSV | 0.281 *** | 0.046 | 6.065 | 0.000 | [0.188, 0.370] | 0.062 [a] | Yes |
| H2a | CII→PUV | 0.096 * | 0.045 | 2.159 | 0.031 | [0.011, 0.187] | 0.008 | Yes |
| H1c | CPI→PHV | 0.594 *** | 0.043 | 13.867 | 0.000 | [0.508, 0.675] | 0.283 [b] | Yes |
| H1d | CPI→PNV | 0.611 *** | 0.042 | 14.690 | 0.000 | [0.529, 0.692] | 0.303 [b] | Yes |
| H1b | CPI→PSV | 0.540 *** | 0.045 | 12.116 | 0.000 | [0.453, 0.626] | 0.227 [b] | Yes |
| H1a | CPI→PUV | 0.719 *** | 0.040 | 17.950 | 0.000 | [0.635, 0.794] | 0.421 [c] | Yes |
| H5 | PHV→WTB | 0.240 *** | 0.051 | 4.673 | 0.000 | [0.138, 0.338] | 0.053 [a] | Yes |
| H6 | PNV→WTB | 0.253 *** | 0.043 | 5.938 | 0.000 | [0.173, 0.341] | 0.066 [a] | Yes |
| H4 | PSV→WTB | 0.145 ** | 0.044 | 3.330 | 0.001 | [0.061, 0.232] | 0.025 [a] | Yes |
| H3 | PUV→WTB | 0.312 *** | 0.048 | 6.537 | 0.000 | [0.217, 0.405] | 0.102 [a] | Yes |
| | Indirect effects | | | | | | | |
| | CPI→PSV→WTB | 0.079 ** | 0.026 | 3.049 | 0.002 | [0.030, 0.132] | | Yes |
| | CPI→PHV→WTB | 0.143 *** | 0.033 | 4.338 | 0.000 | [0.080, 0.209] | | Yes |
| | CPI→PUV→WTB | 0.225 *** | 0.037 | 6.125 | 0.000 | [0.155, 0.298] | | Yes |
| | CPI→PNV→WTB | 0.154 *** | 0.028 | 5.555 | 0.000 | [0.102, 0.211] | | Yes |
| | CII→PHV→WTB | 0.056 ** | 0.017 | 3.246 | 0.001 | [0.025, 0.093] | | Yes |
| | CII→PNV→WTB | 0.054 *** | 0.015 | 3.622 | 0.000 | [0.028, 0.086] | | Yes |
| | CII→PSV→WTB | 0.041 ** | 0.013 | 3.062 | 0.002 | [0.016, 0.069] | | Yes |
| | CII→PUV→WTB | 0.030 * | 0.015 | 1.976 | 0.048 | [0.003, 0.062] | | Yes |

Note: Path significance: *** $p < 0.001$; ** $p < 0.01$; * $p < 0.05$. The levels of significance for the $f^2$ statistic are as follows: [a] > 0.02 (little effect), [b] > 0.15 (moderate effect), and [c] > 0.35 (large effect) [115].

In addition to the $R^2$ values, the size effect $f^2$, is used [115]. There are three different effect values: small (greater than 0.02), medium (greater than 0.15), and large (greater than 0.35) [115]. As seen from Table 6, the $f^2$ value of H1a is equal to 0.421, reflecting the strong influence of consumer product innovation on perceived useful value. The effect of consumer information innovation on perceived useful value is negligible ($f^2 = 0.008$).

In Table 6 and Figure 3, the structural model's validity was evaluated using $R^2$ (a measure of predictive accuracy). The $R^2$ of WTB was 0.797%, indicating that the latent variables explained 79.7% of the purchase intention, which was greater than 50% and demonstrates that the model's assumptions are reasonable. The model fit well [115].

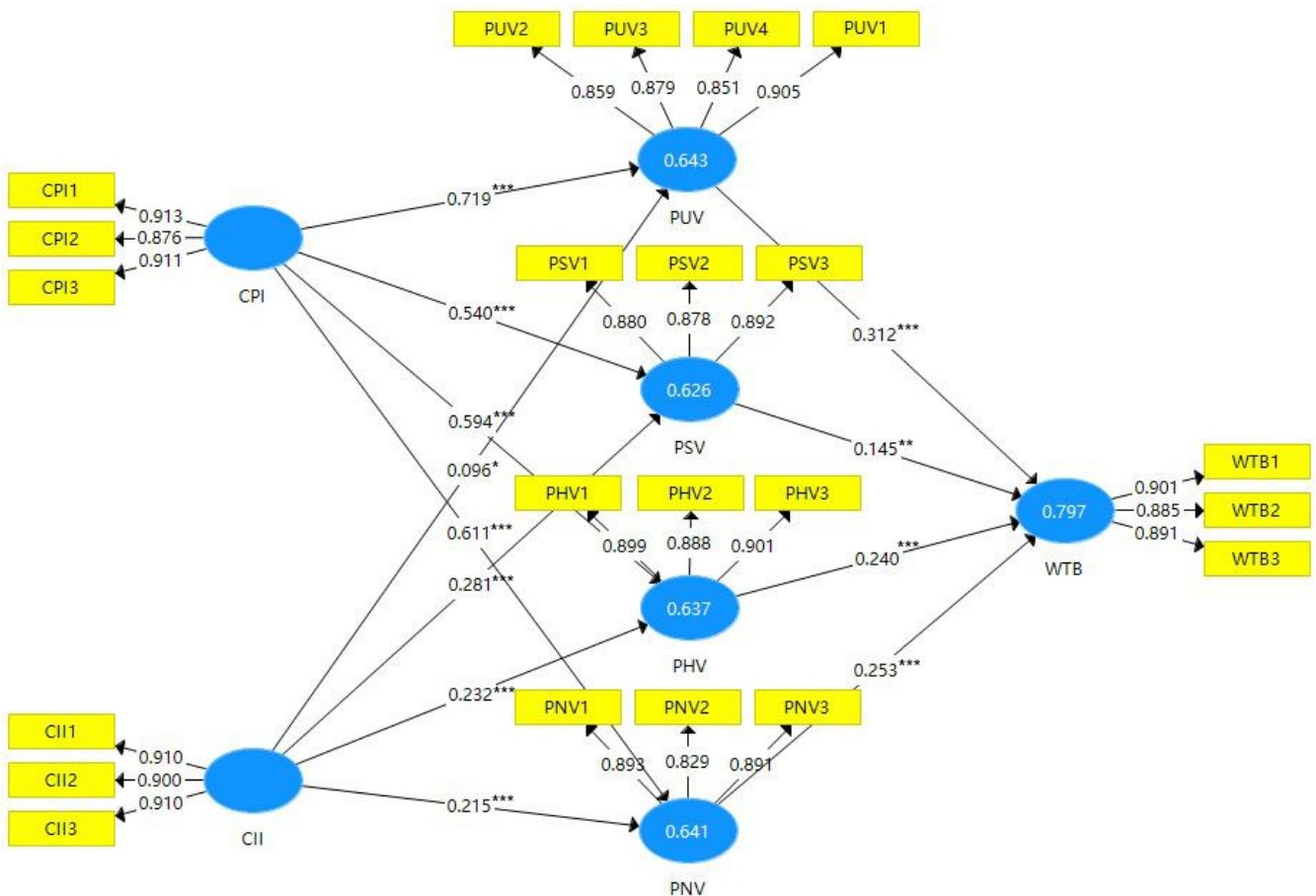

**Figure 3.** Findings of structural model analysis. (Note: *** denote 0.1% significance levels; ** denote 1% significance levels; * denote 5% significance levels.).

From Table 7, it can be concluded that $Q^2 > 0$ indicates that the structural model is valid [124].

**Table 7.** Cv-communality ($Q^2$ for measurement blocks).

| Construct | $Q^2$ |
|---|---|
| PHV | 0.508 |
| PNV | 0.483 |
| PSV | 0.485 |
| PUV | 0.487 |
| WTB | 0.630 |

Note: $Q^2$ = predictive relevance.

According to Wetzels et al. [124], the goodness of Match (GoF) values greater than 0.1 indicate small, 0.25 and 0.36 indicate medium, and GoF values greater than 0.36 indicate largely. The GoF value can be used to examine the model fit, and when the GoF value is higher than 0.36, the model has good applicability [124]. According to GoF's calculation formula: $\text{GoF} = \sqrt{\overline{\text{communality}} \times \overline{R^2}}$ [125]. $\text{GoF} = \sqrt{0.519 \times 0.669} = 0.589$. This model's GoF value was calculated to be 0.589, indicating a decent fit.

4.2.2. Research Model B. (Perceived Value as a Mediator for CPI and WTB; Perceived Value as a Mediator for CII and WTB)

Each path coefficient's statistical significance was evaluated using t-tests, and as noted before, bootstrapping (5000 sub-samples) was utilized to do so. The results are shown in Figure 4.

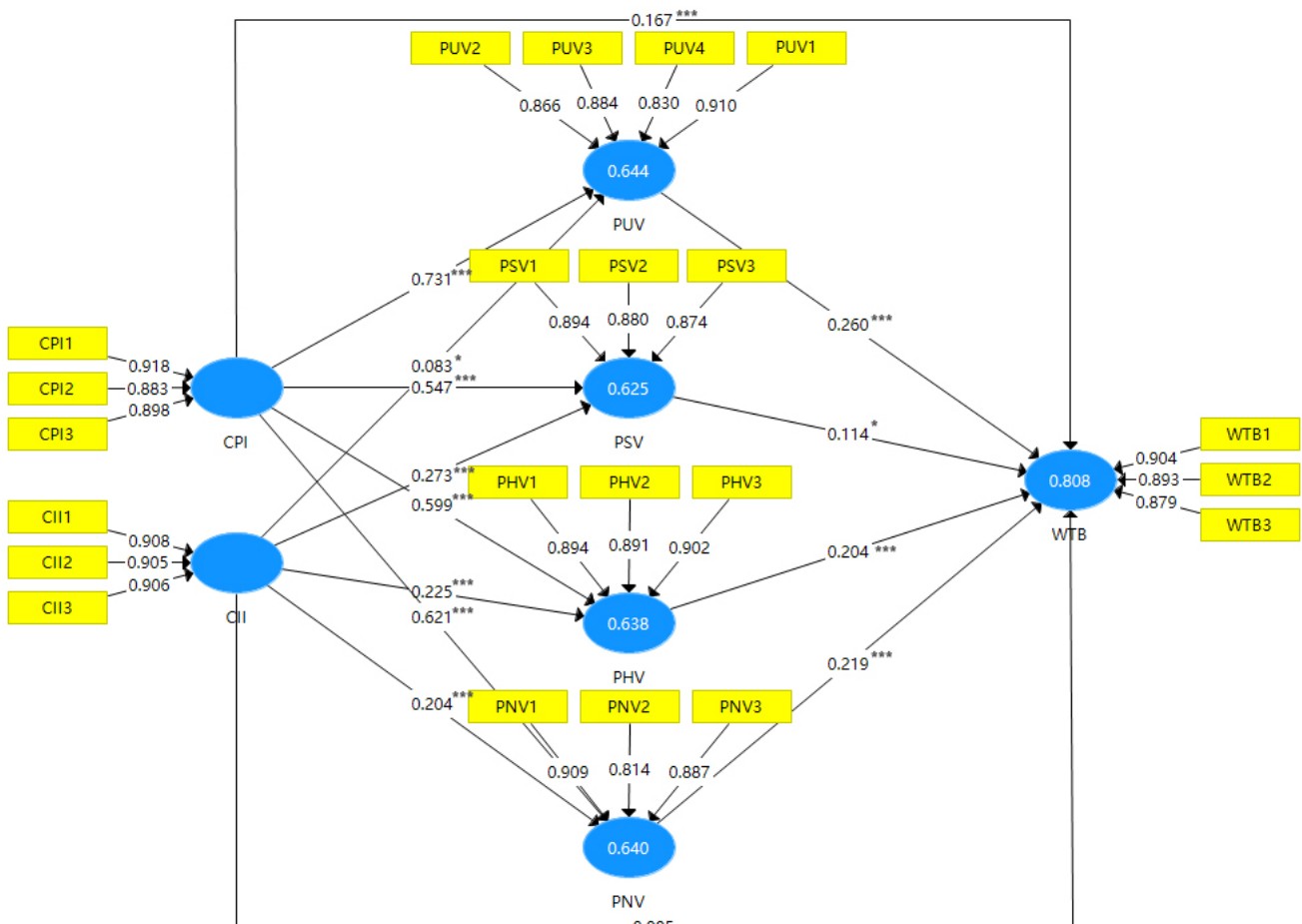

**Figure 4.** Findings of structural model A analysis. (Note: *** denotes 0.1% significance levels; * denotes 5% significance levels.).

Table 7 shows the hypothetical relationships for all effects of study model B.

Analysis results show that the total effect, as well as the CPI of WTB total allow effect, are remarkable. The direct effects of CII on PHV, PNV, PSV, and PUV were significant. The direct effect of PHV, PNV, PSV, and PUV on WTB was significant. The direct effect of CPI on WTB is significant. The indirect effect analysis shows that CPI→PSV→WTB had a significant effect, the indirect effect of CPI→PNV→WTB was significant, the indirect effect of CPI→PUV→WTB was significant, and the indirect effect of CPI→PHV→WTB is significant. Therefore, according to the literature of James, L. and Brett, J. [80], the author will further study whether PHV, PNV, PSV, and PUV have partial mediating effects on CPI and WTB.

Table 8 analyzes the mediating effect of PUV on CPI and WTB, the mediating effect of PSV on CPI and WTB, the mediating effect of PHV on CPI and WTB, and the mediating effect of PNV on CPI and WTB. According to the literature published by Hair et al. [114], it can be known that VAF represents the percentage of indirect effect and total effect. As a rule of thumb, VAF values are divided into three levels: VAF < 20% indicates no mediation effect, 20% ≤ VAF ≤ 80% indicates partial mediation effect, and VAF > 80% indicates complete mediation effect. It can be inferred from the value of VAF that PUV partially mediates CPI and WTB. PSV, PHV, and PNV have no mediating effect on CPI and WTB.

**Table 8.** Hypothesized relationships for all effects (research model B).

| Hypotheses | Path | Estimate | Standard Error | T-Value | *p* Value | 95%CI | $f^2$ | Supported |
|---|---|---|---|---|---|---|---|---|
| | Direct effects | | | | | | | |
| H2c | CII→PHV | 0.231 *** | 0.049 | 4.721 | 0.000 | [0.132; 0.324] | 0.043 [a] | Yes |
| H2d | CII→PNV | 0.215 *** | 0.046 | 4.661 | 0.000 | [0.122, 0.303] | 0.037 [a] | Yes |
| H2b | CII→PSV | 0.281 *** | 0.046 | 6.118 | 0.000 | [0.190, 0.371] | 0.062 [a] | Yes |
| H2a | CII→PUV | 0.096 * | 0.044 | 2.177 | 0.030 | [0.009, 0.180] | 0.008 | Yes |
| H8 | CII→WTB | 0.006 | 0.040 | 0.141 | 0.888 | [-0.073, 0.086] | 0.000 | No |
| H1c | CPI→PHV | 0.594 *** | 0.044 | 13.578 | 0.000 | [0.507, 0.677] | 0.283 [b] | Yes |
| H1d | CPI→PNV | 0.611 *** | 0.042 | 14.650 | 0.000 | [0.529, 0.692] | 0.303 [b] | Yes |
| H1b | CPI→PSV | 0.540 *** | 0.045 | 11.982 | 0.000 | [0.449, 0.624] | 0.227 [b] | Yes |
| H1a | CPI→PUV | 0.719 *** | 0.040 | 17.974 | 0.000 | [0.637, 0.792] | 0.421 [c] | Yes |
| H7 | CPI→WTB | 0.170 *** | 0.046 | 3.677 | 0.000 | [0.083, 0.265] | 0.029 [a] | Yes |
| H5 | PHV→WTB | 0.210 *** | 0.049 | 4.307 | 0.000 | [0.114, 0.302] | 0.041 [a] | Yes |
| H6 | PNV→WTB | 0.211 *** | 0.042 | 4.957 | 0.000 | [0.131, 0.295] | 0.046 [a] | Yes |
| H4 | PSV→WTB | 0.110 * | 0.044 | 2.505 | 0.012 | [0.023, 0.195] | 0.014 | Yes |
| H3 | PUV→WTB | 0.262 *** | 0.048 | 5.416 | 0.000 | [0.164, 0.353] | 0.070 [a] | Yes |
| | Indirect effects | | | | | | | |
| | CPI→PSV→WTB | 0.059 * | 0.025 | 2.411 | 0.016 | [0.013, 0.111] | | Yes |
| | CII→PHV→WTB | 0.049 ** | 0.016 | 3.067 | 0.002 | [0.023, 0.086] | | Yes |
| | CPI→PUV→WTB | 0.188 *** | 0.036 | 5.206 | 0.000 | [0.119, 0.263] | | Yes |
| | CII→PNV→WTB | 0.045 ** | 0.014 | 3.289 | 0.001 | [0.023, 0.077] | | Yes |
| | CII→PSV→WTB | 0.031 * | 0.013 | 2.355 | 0.019 | [0.008, 0.060] | | Yes |
| | CPI→PHV→WTB | 0.125 *** | 0.030 | 4.133 | 0.000 | [0.067, 0.186] | | Yes |
| | CPI→PNV→WTB | 0.129 *** | 0.027 | 4.834 | 0.000 | [0.081, 0.184] | | Yes |
| | CII→PUV→WTB | 0.025 | 0.013 | 1.956 | 0.051 | [0.004, 0.054] | | No |
| | Total effects | | | | | | | |
| | CII→WTB | 0.156 *** | 0.033 | 4.559 | 0.000 | [0.061, 0.252] | | Yes |
| | CPI→WTB | 0.672 *** | 0.036 | 13.833 | 0.000 | [0.581, 0.757] | | Yes |

Note: Path significance: *** $p < 0.001$; ** $p < 0.01$; * $p < 0.05$. The levels of significance for the $f^2$ statistic are as follows: [a] > 0.02 (little effect), [b] > 0.15 (moderate effect), and [c] > 0.35 (large effect) [115].

Table 9 shows that the results of the total effect analysis showed that the total effect of CII on WTB was significant. The direct effect analysis showed that CII significantly

impacted PHV, PNV, and PSV. The direct effect of PHV, PNV, and PSV on WTB was significant. The direct effect of CII on WTB was insignificant. The indirect effect analysis showed that CII→PHV→WTB had a significant indirect effect. The indirect effect of CII→PNV→WTB was significant. The indirect effect of CII→PSV→WTB was significant. Therefore, according to James, L. and Brett, J.'s research [126], the equation is tested by the coefficient c to distinguish between full and partial mediation. If the indirect effects are significant, but the coefficient c is not significant, it belongs to perfect mediation. PHV, PNV, and PSV were judged to mediate CII and WTB fully. The results are shown in Table 10.

**Table 9.** Mediation Tests for Parallel-Sequential Multiple Mediator Models (CPI→WTB).

| Path | Estimate | T-Value | 95%CI | VAF | Final Decision |
|---|---|---|---|---|---|
| H7a: Mediation of PUV (CPI→PUV→WTB) | | | | 28.03% | Partial Mediation |
| CPI→PUV | 0.719 *** | 17.974 | [0.637, 0.792] | | |
| PUV→WTB | 0.262 *** | 5.416 | [0.164, 0.353] | | The hypothesis |
| CPI→PUV→WTB (Indirect effects) | 0.188 *** | 5.206 | [0.119, 0.263] | | Is supported. |
| CPI→WTB (total effects) | 0.672 *** | 13.833 | [0.581, 0.757] | | |
| H7b: Mediation of PSV (CPI→PSV→WTB) | | | | 8.78% | No Mediation |
| CPI→PSV | 0.540 *** | 11.982 | [0.449, 0.624] | | |
| PSV→WTB | 0.110 * | 2.505 | [0.023, 0.195] | | The hypothesis |
| CPI→PSV→WTB (Indirect effects) | 0.059 * | 2.411 | [0.013, 0.111] | | is not supported. |
| CPI→WTB (total effects) | 0.672 *** | 13.833 | [0.581, 0.757] | | |
| H7c: Mediation of PHV (CPI→PHV→WTB) | | | | 18.60% | No Mediation |
| CPI→PHV | 0.594 *** | 13.578 | [0.507, 0.677] | | |
| PHV→WTB | 0.210 *** | 4.307 | [0.114, 0.302] | | The hypothesis |
| CPI→PHV→WTB (Indirect effects) | 0.125 *** | 4.133 | [0.067, 0.186] | | is not supported. |
| CPI→WTB (total effects) | 0.672 *** | 13.833 | [0.581, 0.757] | | |
| H7d: Mediation of PNV (CPI→PNV→WTB) | | | | 19.20% | No Mediation |
| CPI→PNV | 0.611 *** | 14.650 | [0.529, 0.692] | | |
| PNV→WTB | 0.211 *** | 4.957 | [0.131, 0.295] | | The hypothesis |
| CPI→PNV→WTB (Indirect effects) | 0.129 *** | 4.834 | [0.081, 0.184] | | is not supported. |
| CPI→WTB (total effects) | 0.672 *** | 13.833 | [0.581, 0.757] | | |

Note: Path significance: *** $p < 0.001$; * $p < 0.05$.

**Table 10.** Mediation Tests for Parallel-Sequential Multiple Mediator Models (CII→WTB).

| Path | Estimate | T-Value | 95%CI | Supported | Final Decision |
|---|---|---|---|---|---|
| H8a: Mediation of PUV (CII→PUV→WTB) | | | | No | No mediation |
| CII→PUV(a) (direct effects) | 0.096 * | 2.177 | [0.009, 0.180] | | |
| PUV→WTB (b) (direct effects) | 0.262 *** | 5.416 | [0.164, 0.353] | | |
| CII→PUV→WTB (Indirect effects) | 0.025 | 1.956 | [0.004, 0.054] | | |
| CII→WTB (c') (direct effects) | 0.006 | 0.141 | [-0.073, 0.086] | | |
| CII→WTB (c) total effects | 0.156 *** | 4.559 | [0.061, 0.252] | | |
| H8b: Mediation of PSV (CII→PSV→WTB) | | | | Yes | Full mediation |
| CII→PSV (a) (direct effects) | 0.281 *** | 6.118 | [0.190, 0.371] | | |
| PSV→WTB (b) (direct effects) | 0.110 * | 2.505 | [0.023, 0.195] | | |
| CII→PSV→WTB (Indirect effects) | 0.031 * | 2.355 | [0.013, 0.111] | | |
| CII→WTB (c') (direct effects) | 0.006 | 0.141 | [-0.073, 0.086] | | |
| CII→WTB (c) total effects | 0.156 *** | 4.559 | [0.061, 0.252] | | |
| H8c: Mediation of PHV (CII→PHV→WTB) | | | | Yes | Full mediation |

**Table 10.** *Cont.*

| Path | Estimate | T-Value | 95%CI | Supported | Final Decision |
|---|---|---|---|---|---|
| CII→PHV (a) (direct effects) | 0.231 *** | 4.721 | [0.132; 0.324] | | |
| PHV→WTB (b) (direct effects) | 0.210 *** | 4.307 | [0.114, 0.302] | | |
| CII→PHV→WTB (Indirect effects) | 0.049 ** | 3.067 | [0.023, 0.086] | | |
| CII→WTB (c′) (direct effects) | 0.006 | 0.141 | [-0.073, 0.086] | | |
| CII→WTB (c) total effects | 0.156 *** | 4.559 | [0.061, 0.252] | | |
| H8d: Mediation of PNV (CII→PNV→WTB) | | | | Yes | Full mediation |
| CII→PNV(a) (direct effects) | 0.215 *** | 4.661 | [0.122, 0.303] | | |
| PNV→WTB (b) (direct effects) | 0.211 *** | 4.957 | [0.131, 0.295] | | |
| CII→PNV→WTB (Indirect effects) | 0.045 ** | 3.289 | [0.023, 0.077] | | |
| CII→WTB (c′) (direct effects) | 0.006 | 0.141 | [-0.073, 0.086] | | |
| CII→WTB (c) total effects | 0.156 *** | 4.559 | [0.061, 0.252] | | |

Note: Path significance: *** $p < 0.001$; ** $p < 0.01$; * $p < 0.05$.

## 5. Discussion

This study explores the characteristics of consumers who are willing to purchase innovative car seats. Through the study of model A, the hypothesis of this study has been confirmed. Direct effect analysis shows that both consumer product and information innovation have significant effects on perceived value. Truong, Y. Previously found that consumers' innovation ability would positively influence perceived value [109]. The results of this study show that perceived value (perceived product's usefulness, social value, hedonic value, and novelty value) has a significant positive impact on consumers' purchase intention. The previous article has a similar conclusion that the intention to use will be positively affected by perceived usefulness [66,67]. Hedonic value positively impacts consumers' behavioral intentions [74]. Jaleel, A. et al. found that perceived social influence and social value significantly impact usage intention [73]. A recent study by Adapa et al. established a positive correlation between perceived novelty and use intent [77].

In the total effect analysis of this study, consumer product and information innovation positively impact consumers' willingness to buy innovative car seats. This result is consistent with the opinions of articles in other industries. Lee, K. et al. studied whether product innovation significantly impacts the intention to buy smartphone products. The research results show that product innovation significantly positively impacts the intention to buy mobile phone products [127].

By studying model B, the following results are obtained. Perceived product usefulness plays a partial mediating role between consumer product innovation and purchase intention. Perceived social value, hedonic value, and novelty value have no mediating effect on consumer product innovation and purchase intention but only an indirect effect. The impact of consumer product innovation on purchase intent is still much explored in other industries. Saputra M. et al. studied the mediating role of green customer value. The research results show that green customer value has been proven to partially mediate between green product innovation and purchase intention [60].

Model B confirmed the following results. A product's perceived social value, hedonic value, and novelty value fully mediate between consumer information innovation and purchase intention. Researchers are investigating the public acceptance of self-driving cars. Research results show that perceived value fully mediates between consumer innovation and public acceptance of innovative products [128]. Hong et al. [85] proved that hedonic value and utilitarian value play an intermediary role in smartwatch consumers' innovation and use intention.

This study also found that the perceived usefulness of products had no mediating or indirect effect between consumers' information innovation and purchase intention. In the previous study, Abdurrahman C. and Umut A. explored the adoption of smart home

devices [129]. The results show an insignificant relationship between innovation and perceived usefulness in specific domains.

According to the findings above, CPI has a significantly better effect on perceived value than CII [80,130], which means that consumers with a high CPI are more likely to perceive the value of an intelligent safety seat and may be more willing to pay for it than those with a high CII.

### 5.1. Theoretical Contribution

The research model proposed in this study is based on the work of Jeong, S. C. et al. [40,41]. This study further demonstrates that consumers of innovative products and those of innovative information can both positively perceive the novelty and social value of innovative products. The research findings of Jeong, S. C. et al. [40,41] indicate that all of the perceived innovative characteristics of wearable technology positively impact the intention to purchase wearable devices. This study also supports our original assertion that purchasing cutting-edge car seats depends on perceived product value. Furthermore, this study examined the multiple parallel mediating effects of perceived product social value, hedonic value, and novelty value on consumer information innovation and purchase intention of intelligent child safety seats [131]. The analysis also discovered a partial mediating relationship between perceived usefulness and consumers' willingness to purchase innovative child safety seats. The unique and significant theoretical contributions of this study are those mentioned above.

This study defines the DSI structure from consumer product innovation and information innovation. Its relationship with perceived value is examined, followed by the relationship between perceived value and the intent to purchase new products. In prior studies of a similar nature, DSI structure positively influenced the intention to try new products, but the influence coefficient was minimal [84,132,133]. This study introduced TCV structure rather than directly relating DSI structure to purchase intention in model A. This is so because forward-thinking customers will prefer a new product if it can help them achieve their goals or uphold their values rather than just accepting it out of the blue. The findings support our hypothesis that consumers who use innovative products have a favorable impact on perceived value and a strong desire to buy new goods. In light of the evolving business and technological environment, as well as the challenges that contemporary businesses face in terms of individual innovation capabilities, researchers and practitioners need to pay more attention to product-centric consumer innovation capabilities. This result aligns with studies conducted in different fields [134–136].

By examining the impact of perceived value on the intention to purchase smart safety seats, we were able to confirm the importance of the TCV model. This study demonstrates the importance of considering the product's social value, novelty, and alignment with consumers' values, in addition to its features and benefits when studying consumers' propensity to purchase new products [137,138]. Give reasons for and details on how specific users adopt new technologies. It might be a good substitute for the technology acceptance model. Future research may focus on where these two theories on technology adoption converge.

In model B, we investigate the mediating effects of perceived value on both CPI and purchase intention, as well as CII and purchase intention. According to the findings, which are consistent with earlier studies [84,132,133], CPI has a small but significant direct impact on purchase intention. CII had no direct effect on purchase intention, and it is not discussed in the Jeong, S. C. et al. study, nor has it been reported in the literature to date. The results of this study show that perceived product usefulness have a partial mediating effect between CPI and purchase intention, and perceived product social value, hedonic value, and novelty value have multiple parallel mediating effects between CII and purchase intention. These findings suggest that consumers with high CPI may be more inclined to purchase novel products that offer them practical benefits, such as enhancing their work productivity and streamlining the use process. Customers with high CII are more concerned with how well a

new product's novelty, hedonic, and social values align with their own values. Customers with high CII will be more likely to purchase the new product if it matches; otherwise, they may choose not to do so.

Additionally, this study broadens the literature on which this theory is based. This study supported the theory of innovation diffusion in the context of smart safety seats, in contrast to earlier studies that concentrated on the fields of smart wearable technology, online medical applications, and online banking [40,51,139]. We identified the value characteristics of smart safety seats. We confirmed the positive and significant relationship between smart safety seats and purchase intention based on the attributes, consequences, and values of smart safety seats.

Finally, this study confirms the DSI and TCV theories that focus on the intelligent safety seat as the research object. We defined the value of smart seat technology using TCV theory and looked at its relationship to purchasing intent. We also suggest extending the theory by incorporating DSI into this framework. Prior studies have concentrated on the relationship between consumer innovation and adoption behavior [82,140]. Additionally, this study broadens the literature on which this theory is based. This study supported the theory of innovation diffusion in the context of smart safety seats, in contrast to earlier studies that concentrated on the fields of smart wearable technology, online medical applications, and online banking [40,51,139]. We determined the perceived value of innovative safety seats according to the consumer value of innovative safety seats. We verified the significant positive relationship between the perceived value and the purchase intention of innovative safety seats.

Previous research has merged TCV theory with other theories. For example, Dhir et al. (2020) blend the TCV theory with the flow theory and theory of planned behavior. To create comprehensive methods for creating ongoing engagement for mobile instant messaging apps [141].

Carlson et al. (2019) integrated service dominating logic with TCV theory. To investigate how consumer participation in brand communities on the perception of values [142], Wu et al. (2017) coupled the expectation–confirmation theory with the TCV theory to compare the repurchase intention of online versus physical music goods [143].

As far as we know, this is the first paper to meld TCV and DSI perspectives.

Considering the potential contribution and use of TCV in some studies, a hybrid approach must be applied in future TCV studies. These methods provide researchers with flexibility and the ability to apply optimal strategies to answer research questions. Due to the improved effect of quantitative research techniques on research validity, the mixed method ensures the robustness of research results [144]. Therefore, this method can provide more abundant results for researchers to study consumption value from the perspective of pragmatism and is also conducive to studying the consumption value of TCV from the overall perspective. TCV can maintain more adequate results for many consumption value problems with complex relationship structures that need further study. Therefore, hybrid research helps analyze the potential relationship between values and to understand the independence of values. In the future, we will further enrich TCV's existing explanatory capabilities using a hybrid approach to provide detailed, comprehensive, complementary, and holistic knowledge for a deeper understanding of the role of value in consumer behavior.

In the child safety seat industry, there is no previous research article using the partial least squares structural equation model. This study extends the application of PLS-SEM to a new subject area and contributes to the continued development of PLS-SEM.

### 5.2. The Actual Contribution

The study's findings address a gap in consumer research in the market for intelligent car safety seats and provide researchers and marketing companies with concrete recommendations and fixes. The results of this study are anticipated to assist designers and

promoters of intelligent child safety seats in thinking outside the box when formulating new design and marketing strategies.

The study first confirms that, as consumers' assessments of the worth of new products rise, so does their willingness to pay for them. Businesses are therefore compelled to learn more about their target market's ideal outcomes and values to incorporate them into upcoming products. Additionally, sellers must emphasize a new product's accessibility, compatibility with customers' values, and uniqueness when it first enters the market. These can increase consumers' willingness to pay for new goods by boosting their perception of the worth of those goods [145].

In addition, based on the empirical results in the previous section, this study confirms that if the perceived usefulness value of the product is high, then consumers with product innovation characteristics will be more likely to buy innovative car seats. Consumers with information innovation characteristics will have a strong purchase intent if they believe an innovative product has high social value, hedonic value, and novelty. By empirically verifying these propositions, this study significantly contributes to a broader study of the child safety seat industry; This will lead to a better understanding of the various emerging psychological factors influencing consumer behavior toward child safety seats.

This study supports the strategic planning and marketing initiatives of manufacturers, designers, brands, and marketing firms in the innovative child safety seat market. Companies must devise strategies, seek out and communicate with product innovators, ascertain their functional needs, and upgrade products to satisfy those needs to launch innovative car safety seats successfully. In order to better understand the opinions of information innovators regarding the social value, hedonic value, and novelty value of products, businesses can conduct surveys and focus group discussions. They can then attempt to align the brand value with these viewpoints.

It is important to note that this study's conclusions and ramifications do not just apply to the market for innovative car safety seats. We anticipate that more researchers will apply this theory to other industries and further explore and develop the theoretical model to generalize these findings and their implications to other newly developed intelligent product industries.

## 6. Conclusions

Researchers in the car seat industry have been focusing on industry regulations and the abuse of car seats, but there is a lack of consumer-centered research. Therefore, to bridge the gap between the car seat industry and product marketing research, this study explores the influence of consumers with product innovation characteristics and information innovation characteristics on their purchase intentions, respectively. The research shows that consumers with innovative products are more likely to buy new products. In addition, in this research in the car seat industry information on the innovation of the relationship between the consumers and purchase intention, this paper proposes a new perception of the product of social values, hedonic value, and novel parallel multiple mediation relationship of value. Therefore, a parallel multiple mediation model is analyzed using the PLS-SEM method based on variance. In addition, this study emphasizes discovering and understanding the behavior of innovators. Intelligent child safety seat marketers of the future should focus on innovators in the product space, as these people have considerable influence on smart seat purchasing decisions. We built and tested a research model to confirm that consumer innovation positively impacts the perceived value of intelligent safety seats and, in turn, consumers' propensity to purchase these products.

In addition, this study was conducted in a particular cultural context (namely, China), which enjoys a unique identity and prominence in the entire car seat industry as the world's largest producer and exporter of car seats. This research provides significant theoretical, methodological, and contextual contributions to the overall body of knowledge.

As with all studies, this one had some limitations. In order to ensure that the study is rigorous, we took all possible measures to overcome these errors. However, the question-

naire used in this study has the limitations of general self-assessment questionnaires. Future studies can further improve the scale structure by adding supplementary evaluations from others or integrating additional behavioral, psychological, and physical indicators for comprehensive evaluation. In addition, as mentioned above, due to the limited scope of this study, our sample is only from the innovative child safety seats in China, which is only a part of the innovative child safety seats in the world. Therefore, it is recommended that more similar studies be conducted in the child safety seat industry in other countries to verify the claims made in this study in different and broader contexts. The future research direction is to further study the predictability of this model [119] and the relationship between consumer identity and brand [146].

## 7. Patents

The materials used in this research have been patented as follows: [86–106].

**Author Contributions:** Conceptualization, L.J.; methodology, L.J.; software, L.J.; validation, M.Z.; formal analysis, L.J.; investigation, L.J.; resources, H.L.; data curation, L.J.; writing—original draft preparation, L.J.; writing—review and editing, L.J. and L.Y.; visualization, L.J.; supervision, M.Z.; project administration, L.J. All authors have read and agreed to the published version of the manuscript.

**Funding:** This research received no external funding.

**Institutional Review Board Statement:** Not applicable.

**Informed Consent Statement:** Informed consent was obtained from all subjects involved in the study.

**Data Availability Statement:** Not applicable.

**Acknowledgments:** The FORU team's help with the user survey is much appreciated. Many thanks are given to the anonymous reviewers for their thoughtful comments and suggestions. It is greatly appreciated that the editor has been so helpful.

**Conflicts of Interest:** The authors declare no conflict of interest.

## Appendix A

**Table A1.** Measuring Instrument.

| DSI | Derived from |
|---|---|
| Consumer product innovation (CPI)<br>Compared to your peers, you typically own more smart products.You frequently purchase new smart devices before your peers.You typically opt to purchase the newest smart devices. | [63,80,107] |
| Consumer information innovation (CII)<br>You enjoy learning about new information technology.Compared to your contemporaries, you are often more sensitive to knowledge about novel things.You tend to be more interested than your peers in the capabilities and applications of new information technology. | [63,80,107] |
| TCV | |
| Perceived usefulness value (PUV)<br>I think the smart child safety seat would be good for my driving.<br>I could go to my destination more safely if I used the smart child safety seat.<br>The smart child safety seat would make it easier for me to go where I am.<br>I could reach my destination faster if I used the smart child safety seat. | [34] |
| Perceived social value (PSV)<br>I should use smart child safety seats, according to people that matter to me.<br>Those who have the power to affect my behavior believe I should employ smart child safety seats.<br>People whose viewpoints I respect want me to use smart child safety seats. | [34] |

**Table A1.** *Cont.*

| DSI | Derived from |
|---|---|
| Perceived hedonic value (PHV)<br>Using the smart child safety seat would be fun during car trips.<br>Using the smart child safety seat would be enjoyable.<br>Using the smart child safety seat would make me and my kids very happy during car trips. | [108] |
| Perceived novelty value (PNV)<br>The smart child safety seat is a new and refreshing device.<br>Smart child safety seats are unique in this.<br>I think using a smart child safety seat is a novel experience. | [110] |
| Willing to buy (WTB)<br>If I can afford it, I would prefer to buy a smart child safety seat.<br>I intend to use smart child safety seats in the future.<br>I would want to try the smart child safety seat. | [111] |

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
