# Peer review of "How Do Consumer Innovation Characteristics and Consumption Value Shape Users’ Willingness to Buy Innovative Car Safety Seats?"

_sustainability, doi:10.3390/su15010172_

Round 1

Reviewer 1 Report

Dear authors, 

thank you for your manuscript. Please see some comments and suggestions for improvement. 

1.       Please check the grammar/spelling in lines 92-93, 151, 158-162, 268-270, 274, 368

2.       The questionnaire is missing the word at the end of "Perceived usefulness value and" 

3.       Please specify which studies you mean (lines 131, 142, 181)

4.       Missing reference (lines 144, 181-188, 246-247)

5.       Capital letters (lines 150, 179-181, 223-230)

6.       Check the relevance of lines 189-192

7.       Indicate table 1 in the text

8.       Was it approved or it will be (line 269)

9.       Missing author name - the author should be stated (line 266 )

10.   Please provide descriptive statistics 

11.   Check the style of Table 7, ref [67]- lines 391, 394, 418

12.   Please justify the use of items in your questionnaire - Perceived usefulness value. The included items do not actually reflect the goal of this section and specify the particular usefulness of car seats, especially from a child-safety perspective. 

13.   Line 442 mentioned Gender as moderating variable - was it tested in the Result section? please provide the results

14.   Please justify the use of purposive sampling

15.   The operationalization of variables and measurement should be clearer and more concise, with the sources of each scale mentioned.

16.   The willingness as a construct and its mechanism related to TPB has been explored in other studies so that you can provide the background on it, for instance in the article

The enjoyment of knowledge sharing: impact of altruism on tacit knowledge-sharing behavior

How Does Inequality Affect the Residents' Subjective Well-Being: Inequality of Opportunity and Inequality of Effort. Frontiers in psychology, 13, 843854. https://doi.org/10.3389/fpsyg.2022.843854

17.   You may include some articles to provide the industry trends such as  

Vehicle-type strategies for manufacturer's car sharing. Kybernetes, ahead-of-print(ahead-of-print). doi: 10.1108/K-11-2021-1095

18.   There is a major weakness in terms of theoretical support. Please support the hypotheses. Introduce the theoretical support for each, and divide the section 1 by 1. In the end a summary of accepted hypotheses could be done. Some studies that could be useful

Why do consumers prefer a hometown geographical indication brand? Exploring the role of consumer identification with the brand and psychological ownership. International Journal of Consumer Studies, n/a(n/a). doi: https://doi.org/10.1111/ijcs.12806

Understanding the role of influencers on live streaming platforms: when tipping makes the difference. European Journal of Marketing, ahead-of-print(ahead-of-print). doi: 10.1108/EJM-10-2021-0815

Analyzing Intention to Purchase Brand Extension via Brand Attribute Associations: The Mediating and Moderating Role of Emotional Consumer-Brand Relationship and Brand Commitment. Frontiers in psychology, 13, 884673. https://doi.org/10.3389/fpsyg.2022.884673

TBest of luck

Author Response

Response to Reviewer 1 Comments

Dear Editor and reviewers,

Thank you very much for reviewing our manuscript: How do consumer innovation characteristics and value chains shape users' willingness to buy smart seats? And giving such valuable comments.

During the past few days, we have considered these comments carefully and highlighted the corresponding modifications in the revised manuscript with "Track Changes."

We would like to thank the referee again for taking the time to review our manuscript.

Everything goes well.

Your kind consideration will be greatly appreciated.

With best regards,

Sincerely Yours,

authors

Reviewer 2 Report

It is very pleasing to review this manuscript. This manuscript has good logic. And it fills the gaps in the past literature. It has reference value for future researchers. But there are still some minor problems that need to be modified.

1. Page 1. Keywords do not need labels.

2. Line 38. The 2009 literature is inappropriate for describing modern parents.

3. Line 40. "Young parents are increasingly paying attention to the "experience and spirit" of the 40 product, or the emotional experience of the user experience." This sentence should have evidence.

4. Line 45. “Chinese manufacturers have come up with a “new” real product design” Is China the only country that designs intelligent safety seats? Don’t other countries have this design?

5. Line 122. The content and research structure are inconsistent. The research structure shows that CPI and CII have no direct effect on WTB. Also, in this manuscript, there are no statements about mediating effects. Why is the perceived value a mediator variable?

6. Line 168. Reference 38 does not mention that perception is divided into four categories. It is misquoted.

7. Line 194. Reference 43 does not state that product innovation affects hedonic value. And this literature research variable is product innovation not consumer innovation. In addition, the literature cited by the authors is not sufficient to derive research hypotheses.

8. Line 269. It is recommended to use effect instead of impact.

9. Line 269. The research object of this manuscript is not all smart seats. I suggest a modification to the title.

10. Line 505. The lack of relevant data shows why the research model is a non-normal distribution.

11. Figure 3. This manuscript uses SEM. The research model includes binary regression models and multiple regression models. H1 to H2 are binary regression analysis models, and H3 to H6 are multiple regression analysis. Therefore, the hypothesis should include the overall hypothesis and the individual variable hypothesis. It is not consistent with the research structure.

12. Line 368. This manuscript is testing research hypotheses rather than constructing accurate predictive models. So, there is no need to remove any variables from the model. This text is redundant.

13. Table 6. Are there mediator variables in this research structure? There should be relevant discussions.

14. Table 8. This manuscript does not focus on the research topic. The research purpose is to construct a predictive model or to verify the causal relationship of the model.

15. Page 13. Since the authors used SEM. Therefore, the research conclusions should be explained from the view of the SEM model.

Author Response

Response to Reviewer 2 Comments

Thank you very much for reviewing our manuscript: How do consumer innovation characteristics and value chains shape users' willingness to buy smart seats? And giving such valuable comments.

During the past few days, we have considered these comments carefully and highlighted the corresponding modifications in the revised manuscript with "Track Changes."

We would like to thank the referee again for taking the time to review our manuscript.

Everything goes well.

Your kind consideration will be greatly appreciated.

With best regards,

Sincerely Yours,

authors

Reviewer 3 Report

Dear authors,

I appreciate your topic, it is quite original and you conducted a very good research. Still, some issues must be improved. The introduction section could be more specific regarding the actual situation regarding the industry of the car seats. The methodology could describe better the structure of the questionnaire. The discussion section must compare the results to other similar researches.

Al in all the paper is ok. 

Author Response

Response to Reviewer 3 Comments

Dear Editor and reviewers,

Thank you very much for reviewing our manuscript: How do consumer innovation characteristics and value chains shape users' willingness to buy smart seats? And giving such valuable comments.

During the past few days, we have considered these comments carefully and highlighted the corresponding modifications in the revised manuscript with "Track Changes."

We would like to thank the referee again for taking the time to review our manuscript.

Everything goes well.

Your kind consideration will be greatly appreciated.

With best regards,

Sincerely Yours,

authors

Reviewer 4 Report

Please, see the attached document

Author Response

Response to Reviewer 4 Comments

Dear Editor and reviewers,

Thank you very much for reviewing our manuscript: How do consumer innovation characteristics and value chains shape users' willingness to buy smart seats? And giving such valuable comments.

During the past few days, we have considered these comments carefully and highlighted the corresponding modifications in the revised manuscript with "Track Changes."

We would like to thank the referee again for taking the time to review our manuscript.

Everything goes well.

Your kind consideration will be greatly appreciated.

With best regards,

Sincerely Yours,

authors

Round 2

Reviewer 1 Report

Dear authors, 

Thank you for your corrections and adjustments. 

But please check your 3.2. Questionnaire design

The references from section  3.2. Questionnaire design do not match with the references in Appendix A Table A1 Measuring instruments

"The third part is the DSI scale, which is used to measure specific categories of consumer innovation [19, 24, 96, 97]. The fourth part is the questionnaire MEC model. The scale created by the MEC model includes four subscales: perceived usefulness value, perceived social value, perceived hedonic value, and perceived novelty value [34, 98, 99]."

vs References in Appendix 130 - 135? 

Thank you

Reviewer 2 Report

The author made major corrections to my review. But there are still some small problems that need to be corrected.

1. Table 6. There is no corresponding hypothesis for the total indirect effect. It is recommended to delete it. The purpose of this manuscript is not to predict the dependent variable. The total effects are of no significance. Table 8 and Table 9 have the same problem. Remember that the text should also be modified.

2. Figure 4 and Table 6 have inconsistent writing on the level of significance.

Reviewer 4 Report

Please, see the attached document

Round 3

Reviewer 4 Report

The paper has been revised but not all the issues that I raised in my previous report have been addressed. See the comments below:

1. Theory and hypotheses development. Theoretical rationale needs to be explained. Authors need to throw more light on the theory or theories that they have adopted for the study. Besides, the hypotheses development is not well written. The literature review brings in relevant and interesting prior studies, but more time needs to be spent providing the rationale and support for the present research. I think that it is basically a matter of spending more time explaining why the prior literature and theoretical tool provides support for what the authors expect to find. Authors should remember that the literature review better clarifies research needs and explain the research aim and contribution.

2.     The authors should address the discussion section in a proper manner. For instance, the authors should highlight the theoretical and practical contributions of this study. From the study, the theoretical and practical contributions need to be strengthened. Thus, discussions on theoretical contributions need to be improved. For each contribution, authors need to first display what the prior theory says and then argue how their findings extend the theory. Therefore, I would suggest that authors should allow their theoretical depth to guide them to discuss the theoretical and practical/managerial implications section of the research.

Good luck
